# Histone Modifications as Individual-Specific Epigenetic Regulators: Opportunities for Forensic Genetics and Postmortem Analysis

**DOI:** 10.3390/genes16080940

**Published:** 2025-08-07

**Authors:** Sheng Yang, Liqin Chen, Miaofang Lin, Chengwan Shen, Aikebaier Reheman

**Affiliations:** 1Fujian Key Laboratory of Toxicant and Drug Toxicology, Medical College, Ningde Normal University, Ningde 352100, China; 2Fujian Zhengyang Forensic Appraisal Institute, Ningde 352100, China

**Keywords:** histone modification, forensic epigenetics, degraded samples, monozygotic twins, postmortem interval, epigenetic biomarkers, forensic genetics

## Abstract

Histone post-translational modifications (PTMs) have emerged as promising epigenetic biomarkers with increasing forensic relevance. Unlike conventional genetic markers such as short tandem repeats (STRs), histone modifications can offer additional layers of biological information, capturing individual-specific regulatory states and remaining detectable even in degraded forensic samples. This review highlights recent advances in understanding histone PTMs in forensic contexts, focusing on three key domains: analysis of degraded biological evidence, differentiation of monozygotic (MZ) twins, and postmortem interval (PMI) estimation. We summarize experimental findings from human cadavers, animal models, and typical forensic samples including bone, blood, and muscle, illustrating the stability and diagnostic potential of marks such as H3K4me3, H3K27me3, and γ-H2AX. Emerging technologies including CUT&Tag, MALDI imaging, and nanopore-based sequencing offer novel opportunities to profile histone modifications at high resolution and low input. Despite technical challenges, these findings support the feasibility of histone-based biomarkers as complementary tools for forensic identification and temporal analysis. Future work should prioritize methodological standardization, inter-laboratory validation, and integration into forensic workflows. However, the forensic applicability of these modifications remains largely unvalidated, and further studies are required to assess their reliability in casework contexts.

## 1. Introduction

Forensic genetics has traditionally relied on DNA sequence-based markers—most notably short tandem repeats (STRs)—for individual identification, kinship analysis, and crime scene reconstruction [1,2]. While these tools have revolutionized forensic practice, their limitations are increasingly evident in complex scenarios. Severely degraded samples often yield fragmented DNA unsuitable for STR amplification [3]. Monozygotic (MZ) twins, with virtually identical genomes, remain indistinguishable by conventional genotyping approaches [4]. Moreover, static DNA sequences offer no insight into temporal biological changes, limiting their utility in postmortem interval (PMI) estimation [5].

In response to these limitations, epigenetic biomarkers—particularly DNA methylation and histone post-translational modifications (PTMs)—have gained attention as complementary molecular tools in forensic science [6,7,8,9]. While DNA methylation has shown potential in age estimation and tissue identification, its susceptibility to environmental degradation remains a concern in severely compromised samples [6]. In contrast, histone modifications are often embedded within nucleosomes, affording them relatively greater chemical stability and resistance to enzymatic degradation [10,11,12]. These features suggest that histone PTMs may offer more robust signals in forensic contexts involving degraded or environmentally exposed materials [13].

Histone PTMs are dynamic chemical modifications that regulate chromatin architecture and gene expression in a context-dependent manner [14,15]. Several PTMs—including H3K4me3, H3K27me3, and γ-H2AX—have been shown to persist in forensic-type specimens such as bloodstains, bone fragments, and soft tissues [16,17,18]. These modifications also encode environmentally responsive signatures, potentially allowing for discrimination between genetically identical individuals, such as MZ twins [19].

Recent studies using animal models and human cadaver tissues have revealed consistent postmortem kinetics for several histone marks. For example, H3K4me3 and H3K27me3 exhibit tissue-specific stability [20], suggesting a potential for molecular PMI estimation, although direct forensic applications remain largely unvalidated. γ-H2AX displays a biphasic pattern in response to postmortem stress and DNA fragmentation, offering insight into the timing of apoptotic and necrotic processes [21]. In parallel, H3K27ac has shown differential enrichment in MZ twin muscle tissues, supporting its potential role in individual differentiation [22]. Advances in detection techniques—such as CUT&Tag, MALDI imaging mass spectrometry, and nanopore-based epigenomic profiling—have improved sensitivity and resolution in low-input forensic samples [23]. Current findings should be regarded as exploratory and require further forensic validation. While promising, PTM-based forensic applications are still emerging, and studies remain limited in scope and forensic validation. Most studies to date have been conducted in experimental or animal models rather than validated forensic casework, underscoring the gap between conceptual promise and practical implementation.

This review systematically explores the biological significance of histone modifications, their detection strategies, and their emerging applications in forensic science. Special focus is given to three critical domains: degraded sample analysis, MZ twin differentiation, and PMI estimation. By integrating recent case-based evidence and technical innovations, we propose that histone PTMs offer a promising molecular fingerprinting approach for modern forensic genetics. Nevertheless, most current findings originate from non-forensic biomedical research and have yet to be systematically validated under forensic conditions.

## 2. Types and Biological Functions of Histone Modifications

Histone post-translational modifications (PTMs) play essential roles in chromatin regulation and gene expression. Their diverse chemical nature and differential stability render them promising biomarkers in forensic investigations. An overview of key PTM types and their forensic relevance is illustrated in Figure 1.

Collectively, these histone PTMs represent promising molecular signatures for forensic applications, particularly in the analysis of degraded specimens, postmortem interval (PMI) estimation, and monozygotic twin differentiation.

### 2.1. Acetylation

Histone acetylation is one of the most extensively studied post-translational modifications, occurring at the ε-amino groups of lysine residues. This process is catalyzed by histone acetyltransferases (HATs) and reversed by histone deacetylases (HDACs) [24,25]. The addition of acetyl groups neutralizes lysine’s positive charge, weakening histone–DNA interactions and resulting in chromatin relaxation that facilitates transcription [26,27,28]. In contrast, deacetylation reinforces chromatin compaction and represses gene expression [29,30].

In forensic applications, histone acetylation levels correlate with cellular metabolism and environmental stress, which can aid in distinguishing between antemortem and postmortem tissues and assessing sample preservation status [13,31,32]. Jarmasz et al. [13] analyzed human autopsy brain tissues and demonstrated that acetylation marks—including H3K9ac, H3K27ac, H4K5ac, and H4K12ac—remain detectable up to four days postmortem under fluctuating cold and ambient storage conditions, indicating notable short-term PMI stability. These findings suggest that certain acetylation marks could serve as early PMI biomarkers, though interpretation must account for storage variables.

### 2.2. Methylation

Histone methylation occurs on lysine and arginine residues in mono-, di-, or trimethyl forms. Enzymes such as SET1, EZH2, and various PRMTs catalyze methylation, while lysine demethylases (KDMs) reverse it [33,34]. The functional outcome depends on the specific residue methylated: for instance, H3K4me3 and H3K36me3 associate with transcriptionally active chromatin, whereas H3K9me3 and H3K27me3 are linked to repressive heterochromatin [35]. In stem cells, EZH2-mediated H3K27me3 plays a pivotal role in maintaining pluripotency and lineage specification [36].

From a forensic viewpoint, histone methylation embodies a form of “epigenetic memory” that reflects both hereditary and environmental influences. Compared to acetylation, methylation exhibits greater chemical stability, making it especially suitable for highly degraded or archived specimens [37,38]. Borrajo et al. [38]. reported aberrant H3K4me3 patterns in postmortem brain tissue of HIV-infected individuals, hinting at its use in profiling neurodegenerative or drug exposure histories.

### 2.3. Phosphorylation

Histone phosphorylation targets serine, threonine, and tyrosine residues and is mediated by kinases such as ATM, ATR, Aurora B, and MSK1, with reversal by phosphatases including PP1 and PP2A [39,40]. This modification is among the most dynamic and is integral to DNA damage signaling [41], oxidative stress response [42], and cell cycle control [43]. A well-known example is γ-H2AX (H2AX phosphorylated at Ser139), which marks DNA double-strand breaks and is widely used as a genotoxicity biomarker [44,45].

In forensic science, histone phosphorylation may indicate acute cellular stress preceding death. Elevated γ-H2AX levels have been documented in tissues subjected to trauma, hypoxia, or asphyxia, offering potential for distinguishing antemortem events [46,47]. However, due to rapid turnover and environmental sensitivity, phosphorylation marks are best interpreted alongside more stable PTMs like methylation or acetylation [48].

### 2.4. Ubiquitination and SUMOylation

Ubiquitination and SUMO conjugation are common PTMs that regulate chromatin function and genomic stability [49]. Ubiquitination involves covalent attachment of ubiquitin to histone lysine residues—such as H2AK119ub catalyzed by PRC1—contributing to heterochromatin formation and gene silencing [50]. SUMOylation similarly involves modification by small ubiquitin-like modifiers, influencing chromatin compaction and transcriptional repression [51].

These PTMs respond dynamically to cellular stressors such as oxidative damage, viral infection, or thermal exposure [52,53,54]. In forensic contexts, altered ubiquitination or SUMOylation patterns may reflect acute antemortem stress exposures [55,56]. Although direct forensic studies remain limited, emerging high-resolution mass spectrometry offers promise for identifying site-specific “ubiquitin/SUMO fingerprints” for applications in sample integrity evaluation, PMI estimation, and environmental exposure reconstruction [57].

## 3. Techniques for Detecting Histone Modifications

Forensic samples are often characterized by limited quantity, extensive degradation, and contamination with environmental or microbial materials [58,59,60]. Therefore, the detection of histone modifications in such contexts demands highly sensitive, low-input-compatible, and reproducible techniques. Recent technological advances have introduced several promising platforms suitable for these challenging conditions [61,62].

### 3.1. Chromatin Immunoprecipitation Sequencing (ChIP-Seq) and Cleavage Under Targets and Tagmentation (CUT&Tag)

ChIP-seq is a classical technique that employs modification-specific antibodies to immunoprecipitate chromatin fragments, followed by next-generation sequencing to map the genomic distribution of histone marks [63]. Despite its utility, ChIP-seq requires high sample input, complex workflows, and often suffers from elevated background noise, which hampers its application to degraded or trace forensic samples [64].

To address these limitations, Kaya-Okur et al. [65] introduced CUT&Tag, a method that uses antibody-directed Tn5 transposase to simultaneously fragment and tag chromatin at modification sites. This approach enables high-resolution chromatin profiling from as few as 10 cells and has been successfully used to detect H3K4me2 and H3K27me3 in minimal inputs [66]. Its single-cell variant (scCUT&Tag) offers additional benefits in resolution, reproducibility, and signal-to-noise ratio, outperforming earlier single-cell ChIP-seq platforms [67,68]. These immunoenzymatic methods, especially CUT&Tag and its derivatives, represent a major advance in profiling histone PTMs in forensic trace samples.

### 3.2. Mass Spectrometry (MS)

Mass spectrometry (MS) remains the gold standard for characterizing histone post-translational modifications due to its precision, multiplexing capacity, and quantitative resolution [69]. MALDI imaging mass spectrometry (MALDI-IMS) allows for in situ visualization of PTM distributions in tissue samples [70], while LC-MS/MS can resolve combinatorial modifications on histone N-terminal tails [71]. Enrichment strategies such as molecularly imprinted polymers (MIPs) have achieved detection limits as low as 0.5 nM for low-abundance histone peptides, enabling PTM profiling in forensic blood or tissue samples [72].

However, limitations include the high cost, technical complexity, and extensive sample preparation, which may lead to material loss, especially problematic for severely degraded specimens [73]. Enhancements such as antibody-assisted enrichment, tandem mass tags (TMTs), and signal-boosting columns (e.g., BOOST) may increase MS feasibility in forensic settings [74,75].

### 3.3. Emerging Single-Molecule and Multi-Omics Platforms

Emerging platforms now offer single-molecule resolution and integrative epigenetic readouts. NanoHiMe-seq, a nanopore-based long-read sequencing approach, targets histone modifications via engineered methyltransferases to induce local adenine methylation. The resulting methylation pattern is decoded through nanopore sequencing, allowing for concurrent detection of DNA methylation and histone marks at the single-molecule level [76].

This amplification-free method is particularly suited to low-input and degraded forensic samples. In parallel, combinatorial methods such as CUT&Tag-ATAC and single-cell multi-omics are under development, offering simultaneous insights into chromatin accessibility and histone landscapes. These integrated approaches promise to improve the resolution, depth, and interpretability of forensic epigenetic data [77].

In summary, the optimal detection strategy for histone modifications in forensic science depends on the specific constraints of the case, such as sample input, degradation level, and resolution needs. Figure 2 summarizes the key features of representative detection methods currently under evaluation for forensic implementation.

## 4. Challenges in Forensic Analysis of Degraded Samples and the Role of Histone Modifications

### 4.1. Characteristics of Degraded Biological Evidence and Forensic Challenges

In practical forensic investigations, biological evidence is frequently compromised by environmental exposures such as fire, flooding, burial, or advanced decomposition, as well as by long-term storage in historical or archaeological contexts. These conditions often result in extensive DNA fragmentation and chemical alterations—including deamination and oxidative damage—that impair PCR amplification and lead to partial or failed short tandem repeat (STR) profiling, thereby undermining the reliability of genetic identification [60]. In addition, environmental contamination [78] and microbial colonization [79] further complicate downstream analyses, limiting the efficacy of conventional DNA-based methods.

Epigenetic marks, particularly those integrated into chromatin, may demonstrate greater biochemical resilience under such conditions. DNA methylation has already shown partial stability in degraded samples and can retain epigenetic information in certain forensic contexts [80,81]. Histones, as core components of nucleosomes, may offer additional protection by stabilizing associated DNA and preserving regulatory signals even in severely degraded environments [60]. These properties position histone modifications as promising alternative molecular targets for forensic analysis, particularly in cases where DNA integrity is significantly compromised. Thus, the central forensic challenge in degraded specimens lies in recovering meaningful molecular information from structurally damaged materials, a challenge for which histone-based epigenetic profiling may offer a viable solution.

### 4.2. Structural Stability of Histones and Nucleosomes

Nucleosomes, the fundamental units of chromatin, consist of ~147 base pairs of DNA wrapped around histone octamers. This tightly organized structure imparts considerable protection against environmental and enzymatic degradation. Multiple studies have shown that nucleosome-bound DNA is significantly more resistant to degradation than linker regions, highlighting the buffering capacity of the histone–DNA complex under adverse conditions [82,83,84].

Wernig-Zorc et al. [84] employed high-throughput sequencing to compare fresh and degraded biological samples, revealing that DNA fragments preserved post-degradation were disproportionately enriched in nucleosome-occupied regions. This supports the hypothesis that nucleosomes confer structural protection to DNA and chromatin-associated proteins. While direct comparisons between histone and DNA degradation remain limited, the persistent detectability of specific histone marks in postmortem tissues suggests that some histone PTMs may retain relatively stable profiles under forensic conditions [13,85]. Among these, lysine methylation—particularly H3K4me3 and H3K27me3—has emerged as one of the most chemically stable histone marks following death.

Jarmasz et al. [86] demonstrated that DNA and histone methylation levels in postmortem human brain tissue remained largely intact for up to four days after death. Specifically, H3K4me3 levels showed minimal deviation within the first 72 h, closely approximating those in fresh tissues. Conversely, acetylation marks—such as H3K27ac and H3K9ac—were markedly more labile, exhibiting progressive decline under fluctuating storage conditions. Similar observations were reported in animal models, where histone methylation (e.g., H3K27me3) remained stable up to 48–72 h postmortem, while acetylation marks displayed notable degradation beyond 24 h [13]. However, we do not suggest that histone PTMs are universally more stable than DNA sequences. While DNA remains the gold standard for forensic identification due to its robustness and specificity, histone modifications may offer complementary information, particularly in cases where DNA is highly degraded or enzymatically fragmented [60,87]. Their potential lies not in absolute chemical superiority [13,88] but in their utility as proteomic or epigenomic signatures under challenging forensic conditions.

These observations are mainly derived from controlled laboratory studies (e.g., postmortem brain tissues), and their forensic translation remains in its infancy. Collectively, these findings underscore the robustness of histone proteins—particularly methylation marks—as potential biomarkers for forensic analyses involving degraded or temporally aged specimens. Nevertheless, caution must be exercised when interpreting dynamic and environmentally sensitive modifications like acetylation, particularly in relation to the postmortem interval and sample storage conditions.

### 4.3. Forensic Applications of Histone Modifications in Degraded Samples

Although preliminary, given their structural resilience and biochemical properties, histone modifications offer several promising avenues for forensic applications in the context of degraded biological evidence.

#### 4.3.1. Epigenetic Fingerprinting for Individual Identification

Although histone modifications are dynamic and responsive to environmental stimuli, certain patterns may exhibit inter-individual variability in specific tissues or pathological conditions. For example, H3K27me3 displays tissue-specific distributions in the liver and myocardium with reproducible individual-level differences [12,89]. In cases where genomic DNA is insufficient for STR or SNP profiling, these epigenetic patterns may serve as indirect “epigenetic fingerprints” for personal identification [20,90]. While still conceptual, this approach could be validated through large-scale population datasets and tissue-specific epigenomic mapping.

#### 4.3.2. Molecular Estimation of Sample Degradation

The relative stability of different histone marks may serve as an internal molecular clock to assess sample degradation status or approximate postmortem interval (PMI). For instance, the signal intensity ratio between a stable methylation mark (e.g., H3K4me3) and a labile acetylation mark (e.g., H3K9ac) may reflect the temporal degradation stage. Preliminary models have attempted to mathematically characterize the decay kinetics of these modifications, providing an objective, molecular-level assessment of sample quality independent of morphological or environmental indicators [91,92].

#### 4.3.3. Expanded Analysis of Challenging Substrates

Histone modifications may extend molecular profiling capabilities to forensic substrates traditionally considered unsuitable for DNA analysis, such as bone, hair shafts, enamel, or mummified tissues [93,94]. Bone tissue, in particular, retains nucleosomal architecture in marrow remnants even under advanced decomposition. Profiling histone PTMs in such matrices could significantly broaden the scope of forensic epigenetics [95]. A 2024 review by Procopio and Bonicelli [16] reported viable epigenetic and proteomic signatures in “DNA-depleted” bone samples, underscoring the feasibility of histone modification analysis in historical and skeletal remains.

#### 4.3.4. Insights into Cause of Death and Forensic Pathology

In addition to identity and degradation analysis, histone modifications can provide mechanistic insights into cellular stress and death pathways, supporting forensic pathology. For instance, H4K20me3 is involved in DNA damage response and has been reported to vary between violent and natural death scenarios [96,97]. Likewise, hypoxia-induced stress is associated with chromatin condensation and global histone deacetylation, potentially producing characteristic epigenetic imprints [95]. In cases such as sudden cardiac death, abnormal histone modification patterns in myocardial tissue (e.g., H3K9me2 and H3K27ac) may serve as molecular indicators of stress-induced cardiac events [98,99]. While this area remains exploratory, it holds promise for integrating histone-based biomarkers into routine forensic autopsy workflows. Table 1 summarizes key histone modifications with putative forensic utility across various biological contexts. To date, no standardized workflows or reference databases exist for histone PTM interpretation in forensic applications, making their practical utility speculative.

## 5. The Forensic Potential of Histone Modifications in Differentiating Monozygotic Twins

### 5.1. Challenges and Emerging Strategies in Monozygotic Twin Identification

Monozygotic (MZ) twins originate from a single zygote and share nearly identical nuclear DNA sequences, rendering them indistinguishable by standard forensic genotyping methods such as STR or SNP analysis [103,104]. This poses a significant challenge in criminal investigations or kinship analyses involving MZ individuals. Two main strategies have emerged to address this limitation:(1)Somatic Mutation Profiling via High-Throughput Sequencing

MZ twins accumulate somatic mutations post-zygotically, including SNVs and indels, which can be identified by whole-genome (WGS) or whole-exome sequencing (WES) [105,106]. Case reports have demonstrated that such discordant variants can differentiate MZ individuals [107,108]. However, forensic application remains limited by high costs, technical complexity, tissue-specific variability, and uncertain temporal stability [109,110].

(2)Epigenetic Divergence as a Forensic Tool

Epigenetic marks, including histone modifications, diverge progressively between MZ twins due to differential environmental exposures, lifestyle, and disease history. Compared to somatic mutations, such divergence is more widespread and dynamic [111,112]. These accumulating differences—while less stable than DNA variants—may serve as forensic biomarkers for MZ twin discrimination, especially when integrated with advanced epigenomic profiling technologies.

### 5.2. Evidence of Histone Modification Differences Between Monozygotic Twins

(1)Tissue-Specific Divergence

Histone modifications, such as H3K4me3 and H3K27me3, exhibit strong tissue specificity [113,114]. Longitudinal studies have shown that epigenomic divergence between MZ twins increases with age and environmental heterogeneity [112,115,116,117]. ChIP-seq analyses of the prefrontal cortex and cingulate gyrus—regions linked to cognition—have revealed notable differences in histone methylation between twins [118]. These variations may reflect differential cognitive stimulation, stress, or disease burden [119]. Although direct brain sampling is unfeasible in forensic practice, peripheral tissues like blood or buccal cells may carry correlative epigenetic patterns [120], offering viable alternatives for forensic inference.

(2)Chromatin Accessibility and Active Histone Marks

Active histone marks such as H3K4me3, H3K27ac, and H3K9ac correlate with gene expression and open chromatin states [100]. Environmental exposures can alter these marks. For instance, smoking has been linked to elevated H3K27ac in leukocytes, potentially reflecting enhanced transcription of inflammatory genes [121,122]. Stress, dietary habits, and physical activity may similarly modulate acetylation levels, offering avenues for capturing lifestyle-associated epigenetic divergence between twins.

(3)Immune and Disease-Related Epigenetic Remodeling

Repressive marks like H3K9me3 and H3K27me3 govern immune gene silencing. Disease-specific histone modifications have been observed in discordant MZ twin pairs, especially in immune cells [118,123,124]. For example, altered repressive marks in PBMCs may indicate chronic inflammation or autoimmune pathology in one twin but not the other. Such histone signatures could support twin discrimination in contexts where one twin exhibits subclinical or overt disease phenotypes.

Overall, although histone modifications are more dynamic than DNA sequence variants, their abundance and responsiveness to environmental factors make them attractive epigenetic biomarkers for MZ twin differentiation. While current efforts have focused largely on DNA methylation markers (e.g., cg12597325 and cg01095518) [124], histone-based profiling remains a promising and underexplored frontier. With the integration of single-cell epigenomics, machine learning, and high-resolution profiling, it is anticipated that a discriminative panel of histone marks may be developed to complement conventional DNA-based approaches in resolving forensic cases involving MZ twins. An overview of histone modification divergence in monozygotic twins is presented in Figure 3.

Environmental exposures such as smoking can alter the histone modification landscape in genetically identical individuals. Peripheral blood, buccal cells, and brain tissue are potential sources for chromatin extraction. Selected histone marks, including H3K4me3 (active promoter) and H3K27ac (active enhancer), are profiled using the CUT&Tag technique involving antibody targeting, Tn5 transposase integration, PCR amplification, and library preparation. The resulting chromatin profiles are visualized as a heatmap (representative only) to illustrate inter-individual differences in histone modification patterns. These differences may enable individual discrimination in monozygotic twins for forensic applications.

## 6. Histone Modifications and Postmortem Interval (PMI) Estimation

### 6.1. Current Status and Challenges in PMI Estimation

Accurately estimating the postmortem interval (PMI) is a critical task in forensic science, essential for reconstructing the timeline of death, identifying suspects, and determining the cause of death [125]. Conventional PMI estimation primarily relies on physiological and morphological changes, including algor mortis (cooling of the body), livor mortis (settling of blood), and rigor mortis (muscle stiffening and relaxation) [126]. Additional indicators, such as vitreous humor potassium concentration [127], gastric content digestion [128], and entomological succession [129], may also assist. While these methods are useful in the early postmortem period (within a few hours) or under stable environmental conditions, they often suffer from significant variability and individual differences. Especially during the mid-to-late PMI (>2–3 days), environmental factors such as temperature, humidity, and burial conditions can greatly affect decomposition, limiting the applicability of traditional empirical models.

In recent years, molecular degradation markers such as RNA half-lives and protein degradation rates have gaining increasing attention PMI indicators. However, RNA is highly susceptible to degradation and easily influenced by infection and inflammation [130], while protein-based approaches face challenges related to tissue-specific variability and strict sample handling requirements [131]. Currently, there remains a lack of endogenous molecular markers that are both time sensitive and broadly applicable. This has prompted exploration into epigenetic markers. If certain histone modifications exhibit reproducible, time-dependent patterns after death (e.g., linear increase or decrease with PMI), they could serve as endogenous “molecular clocks,” offering more objective and accurate tools for long-range PMI estimation.

### 6.2. Postmortem Dynamics of Histone Modifications

Death is not a sudden event at the cellular level but a progressive biological process involving metabolic arrest, chromatin remodeling, protein degradation, and immune activation [132,133]. Histone modifications, as central regulators of chromatin structure, undergo time-dependent changes during this process. These changes reflect both the cessation of transcriptional activity (e.g., decrease in active marks and relative increase in repressive marks) [134,135] and the influence of residual enzymatic activity [136] and environmental factors [137]. Although limited, several studies in animals and humans have investigated these postmortem trajectories.

#### 6.2.1. Stability and Changes in the Short-Term PMI

Early work by Huang et al. [18] compared chromatin status in postmortem human brains and found that core histones remained tightly associated with DNA within 30 h after death. Active marks such as H3K4me3 and repressive marks like H3K27me3 at specific loci were similar to those in fresh samples. Global H3 methylation levels also remained unchanged between 5 and 30 h postmortem and across typical pH ranges (6.0–6.8). Similarly, Jarmasz et al. [13] reported that H3K4me3 in the human prefrontal cortex could still be reliably detected up to 72 h postmortem, with no significant signal decline. These findings suggest that some histone modifications remain detectable during the early postmortem period, indicating potential utility for short-term PMI estimation; however, systematic forensic validation is lacking. Furthermore, studies have shown that overall activity of histone acetyltransferases and methyltransferases does not significantly change with increasing PMI or prolonged storage [138]. This implies that existing histone marks persist while enzymatic activity ceases, creating a relatively static “epigenetic relic.”

#### 6.2.2. Degradation Patterns in the Mid-to-Long-Term PMI

As PMI lengthens, histone proteins and their modifications inevitably decay. H3K27-related marks, in particular, exhibit rapid degradation. In Sprague–Dawley rats, levels of H3K27me3 and H3K27ac declined rapidly as PMI extended, showing less stability than DNA methylation [12]. This may be due to proteolytic degradation of exposed histone N-terminal tails, where most functional modifications are located. Autolysis and putrefaction-induced changes in tissue pH and ion balance may also affect histone-modifying enzymes [138]. Clinical epigenetics reported that in postmortem brain tissue, acetylation marks like H3K9ac and H4K5ac showed noticeable decline within 48–72 h, highlighting their vulnerability [13]. Most current studies focus on PMI ranges of 1–5 days, while systematic data on histone dynamics beyond several weeks are lacking [18,139]. Ethical and logistical constraints limit human studies, making it difficult to establish long-term quantitative relationships between histone changes and PMI.

Nevertheless, the relatively slow degradation of nucleosome-associated chromatin particularly in deep tissue contexts—and the selective persistence of certain methylation marks under controlled conditions—suggest that some histone PTMs may retain forensic value beyond the acute phase of decomposition. Further research using animal models with longer postmortem intervals is needed to determine whether specific marks (e.g., H3K4me3 or H3K9me2) can serve as molecular indicators in advanced decomposition stages where DNA or RNA may already be extensively degraded.

#### 6.2.3. Tissue-Specific Variation in Histone Decay

Histone modification degradation rates vary between tissues due to differences in enzymatic content, microbial activity, and environmental exposure [140]. Brain tissue, protected by the skull and with lower enzymatic activity, tends to preserve epigenetic marks longer [141]. In contrast, the liver, rich in enzymes, may exhibit faster loss of modifications. Forensic scientists must compare tissues like muscle, liver, and brain to identify optimal sampling sites and histone markers for PMI inference [142,143]. For instance, comparative studies on brain and blood miRNAs for PMI estimation revealed different degradation kinetics [144,145]. Similarly, certain H3 modifications may remain detectable for 96 h in the brain but only 48 h in the liver. Such cross-tissue comparisons can inform evidence-based sampling strategies tailored to various death scenarios.

### 6.3. Application Prospects and Future Directions

Future research should focus on (1) constructing time-resolved degradation curves of histone modifications using animal models; (2) simulating postmortem conditions (e.g., temperature, pH, and hypoxia) in vitro to investigate modification stability; (3) collecting postmortem human samples across diverse PMI intervals for cross-sectional validation; and (4) integrating histone modifications with other omics indicators (e.g., DNA methylation, mRNA stability, and proteomic profiles) to build robust multiparametric PMI models. Only through such efforts can histone modifications be transformed into reliable, quantifiable, and reproducible forensic biomarkers for PMI estimation.

In summary, dynamic postmortem changes in histone modifications hold promise as molecular clocks for PMI inference. However, significant research is still needed before clinical or forensic application. Key questions remain: (1) Which modifications exhibit the most consistent and PMI-correlated changes? (2) Are these patterns robust across environmental conditions? (3) How can information from multiple tissues and histone marks be integrated into accurate models? Encouragingly, with the integration of multi-omics and artificial intelligence into forensic research, several studies have already shown improved PMI prediction using machine learning models trained on proteomics and metabolomics data [10,146]. For example, a 2024 study introduced a multi-omics stacking model (MOSM) that combined metabolic, proteomic, and infrared spectral data, achieving a PMI prediction accuracy of 0.93 and an AUC of 0.98 [147]. This suggests that a similar approach could be applied to histone modifications, using machine learning to construct composite models and enhance PMI estimation accuracy. With continued data accumulation and refinement of analytical tools, histone-based PMI inference may soon become a practical and scientific complement to traditional methods. Postmortem trends of selected histone marks across PMI stages are schematically presented in Figure 4. It remains to be seen whether histone PTMs can complement or outperform traditional PMI markers in real-world forensic scenarios, particularly beyond short postmortem intervals.

## 7. Emerging Case-Relevant Evidence and Forensic Implementation Prospects

Although histone modification analysis has not yet been formally introduced as admissible evidence in judicial proceedings, growing experimental data support its applicability to forensic casework [148]. This section summarizes current studies conducted on postmortem human tissues, degraded skeletal remains, and animal models that simulate forensic conditions, offering a foundation for translational application.

### 7.1. Histone Modifications in Postmortem Human Tissues

Several studies have demonstrated the detectability and stability of histone modifications in human autopsy tissues [13,149,150]. Jarmasz et al. [13] showed that acetylation and methylation marks—such as H3K9ac, H3K27ac, and H3K4me3—remain detectable for up to 72 h postmortem in brain samples stored under fluctuating environmental conditions. These findings underscore the biochemical resilience of certain histone marks and support their use in short-term postmortem analyses, such as PMI estimation and sample degradation assessment.

### 7.2. Degraded Bone and Skeletal Remains

Forensic contexts often involve highly degraded remains, including skeletal fragments. Procopio and Bonicelli [16] reported that histone PTMs are detectable in ancient and DNA-depleted bone samples using proteomics and epigenomic profiling. These findings suggest that nucleosome-associated histone marks may survive in harsh environmental conditions and could be applied in the analysis of archaeological or unidentified skeletal remains, where traditional DNA profiling is no longer feasible.

### 7.3. PMI Modeling in Animal Studies

Controlled animal studies have provided preliminary models of postmortem histone dynamics. Huang et al. [6] and Dulka et al. [85] investigated the time-dependent degradation of H3K27me3 and γ-H2AX in rodent brain tissues and found consistent patterns of signal decay across defined PMI intervals. These models provide empirical evidence for the development of histone-based molecular clocks and could be further validated in forensic autopsy cohorts.

### 7.4. Standardization and Validation Pathways

The integration of histone PTMs into forensic practice requires the development of standardized workflows and interpretive frameworks. Efforts have begun to adapt histone profiling techniques—such as CUT&Tag [65] and MALDI-IMS [151]—to forensic-compatible formats, including FFPE tissues, dried blood spots, and low-input bone marrow samples. Simulated casework, blind trials, and cross-laboratory validations will be essential to establish reproducibility and judicial admissibility [152,153]. Collaborations between forensic institutions and molecular epigenetics laboratories are urgently needed to accelerate this translational process.

Nonetheless, current studies are exploratory and have yet to demonstrate validated forensic applicability. Future work should focus on building reference datasets, optimizing protocols, and establishing cross-laboratory reproducibility. With continued method refinement and validation, histone-based analysis may become a viable supplement to conventional forensic techniques.

## 8. Conclusions and Future Perspectives

Despite these theoretical advantages, no consensus protocols or forensic validation studies currently exist for routine casework implementation of histone PTMs. Taken together, this review synthesizes converging evidence that histone post-translational modifications (PTMs)—particularly methylation (e.g., H3K4me3 and H3K27me3) and phosphorylation (γ-H2AX)—offer a novel class of epigenetic biomarkers for forensic applications, addressing three long-standing limitations: degraded sample analysis, monozygotic twin differentiation, and postmortem interval (PMI) estimation [6,17,20]. However, translating these molecular signatures into forensic practice requires overcoming interdisciplinary barriers spanning technical, interpretative, and ethical domains.

### 8.1. Technical Limitations and Solutions

Current gold-standard methods, such as ChIP-seq, require large cell numbers (>10,000 cells), limiting their forensic application [63,64]. Single-cell CUT&Tag protocols reduce input demands to as few as 10 cells, yet optimization for FFPE tissues and dried bloodstains remains necessary [65,66,154,155]. Furthermore, heterogeneous forensic samples (e.g., bone, muscle, and hair shafts) require tissue-specific protocols [16,93]. A forensic histone consortium could facilitate standardized validation across platforms, such as PAT-ChIP for archived samples [156,157]. Additionally, postmortem stability varies significantly among histone modifications (e.g., H3K4me3 stable for 72 h; H3K9ac rapidly degrades) [13,85]. Developing kinetic decay models that integrate environmental parameters (temperature and pH) could address these stability concerns [91,92]. These limitations, proposed solutions, and implementation challenges, including costs, are briefly summarized in Table 2.

### 8.2. Data Interpretation and Judicial Admissibility

Interpreting histone PTM patterns necessitates distinguishing biological variation from technical noise [158]. Machine learning (e.g., random forests and neural networks) trained on large-scale reference datasets (e.g., ENCODE or GTEx histone profiles) can generate probability-based discrimination thresholds [159,160]. For courtroom admissibility, blind trials mimicking forensic casework (e.g., twin differentiation using H3K27ac [100,161]) must demonstrate reproducibility across laboratories [162].

### 8.3. Ethical and Privacy Considerations

Histone PTMs may reveal sensitive health information (e.g., H3K27ac linked to smoking [121]). We propose the following: minimal data collection: limit profiling to forensically relevant loci (e.g., 20 high-variability marks) [163]; anonymization by design: encrypt epigenetic data linked to identifiers [164]; and judicial oversight: require court orders to access health-relevant PTMs [165].

### 8.4. Future Directions

We suggest the following: short-term (1–3 years): cross-laboratory validation of five core marks (H3K4me3, H3K27me3, H3K9ac, H4K20me3, and γ-H2AX) in standardized degraded samples [72,75]; mid-term (5 years): integrate with multi-omics PMI models (e.g., combine H3K27me3 decay with mitochondrial DNA degradation [147]); and long-term (10 years): develop portable, antibody-free kits (e.g., nanopore sequencing of PTMs [76]).

### 8.5. Concluding Remarks

Histone epigenetics is poised to transform forensic science, offering solutions to previously intractable challenges, from identifying disaster victims to exonerating the innocent. Realizing this potential demands collaboration among molecular biologists, computational scientists, and legal experts, ultimately advancing justice through the lens of chromatin biology.

## Figures and Tables

**Figure 1 genes-16-00940-f001:**
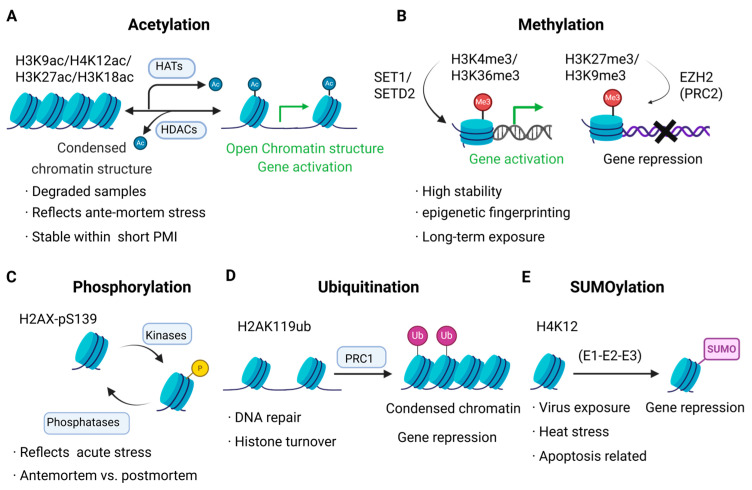
Representative histone post-translational modifications (PTMs) and their forensic significance. (**A**) Acetylation of lysine residues (e.g., H3K9ac and H4K12ac), catalyzed by histone acetyltransferases (HATs) and reversed by histone deacetylases (HDACs), promotes chromatin relaxation and transcriptional activation. In forensic contexts, acetylation marks remain detectable during short postmortem intervals and may reflect antemortem physiological stress in degraded samples. (**B**) Methylation at lysine residues (e.g., H3K4me3 and H3K27me3) is mediated by methyltransferases such as SET1/SETD2 (activation) and EZH2 within the PRC2 complex (repression). These modifications are chemically stable and useful for epigenetic fingerprinting and retrospective exposure assessment. (**C**) Phosphorylation, exemplified by γ-H2AX (H2AX-pS139), is dynamically regulated by kinases and phosphatases in response to DNA damage and oxidative stress, offering a potential marker for distinguishing antemortem and postmortem cellular states. (**D**) Ubiquitination (e.g., H2AK119ub), catalyzed by the PRC1 complex, facilitates chromatin compaction and transcriptional repression and is associated with DNA repair and histone degradation pathways. (**E**) SUMOylation, including modifications such as H4K12-SUMO1, is executed via E1–E2–E3 enzyme cascades and contributes to gene silencing under conditions of viral infection, heat stress, or programmed cell death.

**Figure 2 genes-16-00940-f002:**
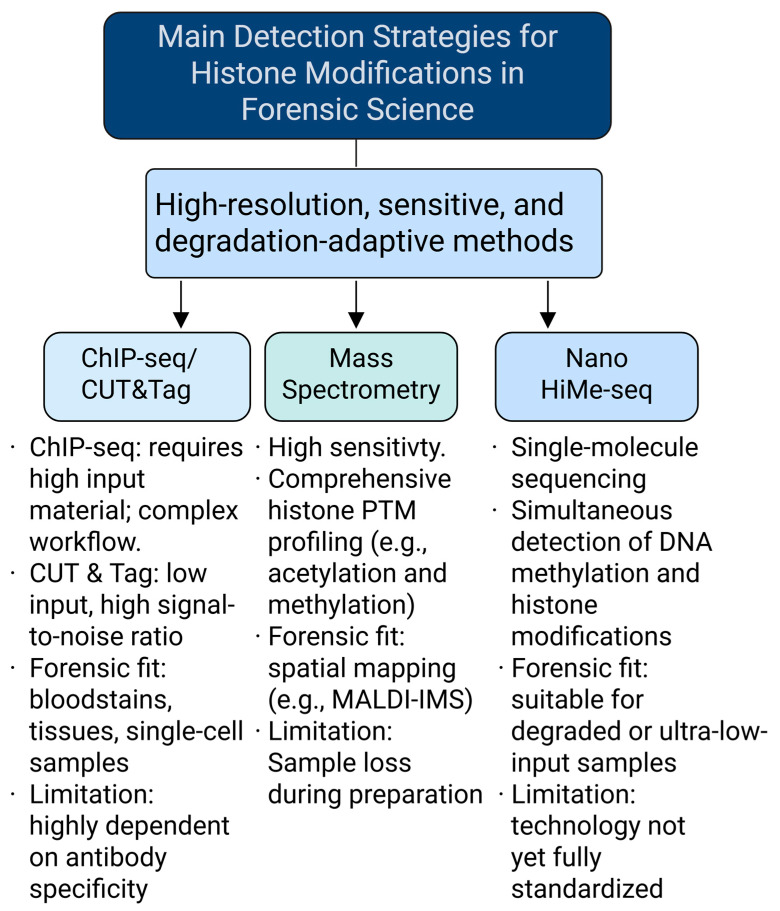
Representative detection strategies for histone modifications in forensic science. Three major approaches are currently employed to profile histone post-translational modifications (PTMs) in forensic applications. (1) Chromatin immunoprecipitation sequencing (ChIP-seq) and Cleavage Under Targets and Tagmentation (CUT&Tag) utilize antibody-mediated enrichment of chromatin fragments. While ChIP-seq requires substantial input material and involves a labor-intensive workflow, CUT&Tag is optimized for low-input or trace samples, offering improved signal-to-noise ratios and compatibility with single-cell resolution. Both methods are applicable to bloodstains, tissue samples, and degraded forensic specimens but remain limited by antibody specificity and batch variability. (2) Mass spectrometry (MS) provides high sensitivity and multiplexed analysis of histone PTMs, including acetylation, methylation, and phosphorylation. Techniques such as MALDI imaging mass spectrometry (MALDI-IMS) enable spatial localization of PTMs within tissue sections. However, MS workflows are often constrained by extensive sample preparation, high instrumentation costs, and the risk of sample loss. (3) NanoHiMe-seq, a nanopore-based single-molecule sequencing technology, allows for simultaneous detection of histone modifications and DNA methylation without requiring amplification. It is particularly suitable for ultra-degraded or low-input forensic samples but has yet to achieve standardization for routine casework implementation. Collectively, these techniques represent a significant advance toward high-resolution, sensitive, and degradation-resilient epigenetic profiling in forensic science.

**Figure 3 genes-16-00940-f003:**
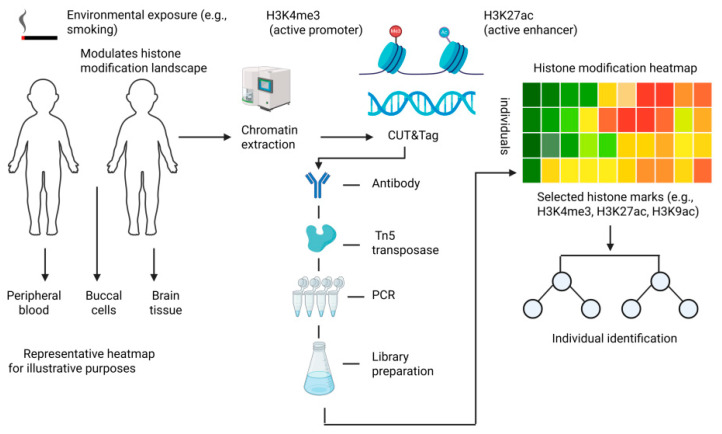
Epigenetic divergence in monozygotic twins and its forensic potential revealed by histone modification profiling.

**Figure 4 genes-16-00940-f004:**
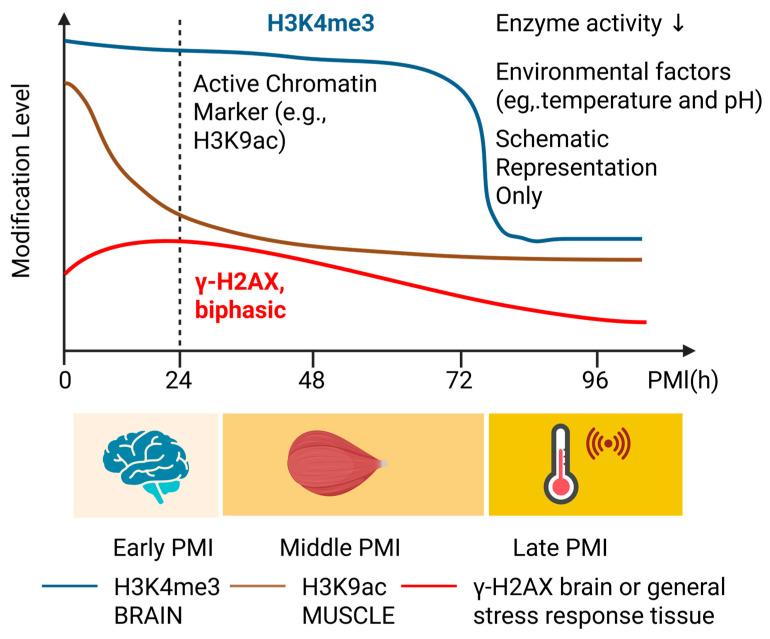
Schematic representation of histone modification dynamics over postmortem intervals (PMI). H3K4me3 (blue), predominantly found in brain tissue, demonstrates high stability up to 72 h postmortem. H3K9ac and H4K5ac (brown), markers of active chromatin in muscle, rapidly decline due to enzymatic degradation. γ-H2AX (red), a DNA damage-associated marker, shows a hypothetical biphasic trend, potentially reflecting early perimortem stress followed by degradation. Enzyme inactivation and environmental conditions (e.g., temperature and pH) influence these postmortem trajectories. This schematic is based on published literature trends and not derived from experimental measurements.

**Table 1 genes-16-00940-t001:** Forensic potential and research value of histone modifications across different application scenarios.

Forensic Application	Challenges	Advantages of Histone Modifications	Representative Studies
Analysis of Degraded Biological Samples	Severe DNA fragmentation, STR amplification failure, and susceptibility to environmental contamination and microbial interference, leading to failure of conventional genotyping methods	Histones coexist with DNA in chromatin and are protected by stable nucleosomal structures; certain histone modification sites show high stability and can serve as alternative molecular markers	Acetylation marks such as H3K9ac, which reflect cellular activity and degradation status, were reported by Jarmasz et al. [13] to remain detectable up to 72 h postmortem, indicating short-term stability.
Monozygotic Twin Individual Identification	Identical nuclear genomic sequences; conventional STR genotyping fails to distinguish between genetically identical individuals	Epigenetic modifications are influenced by environment, behavior, disease, and lifestyle, leading to inter-individual variation with temporal and tissue specificity; useful for assessing exposure history and physiological status	Histone marks such as H3K4me3 and H3K27me3 show stable differences between monozygotic twins [100,101]. ChIP-seq has been successfully applied to detect such differences in peripheral blood samples [63,64].
Postmortem Interval (PMI) Estimation	Conventional indicators have large margins of error; individual variation and environmental influences complicate decomposition rate and PMI estimation	Certain histone modifications (e.g., methylation and phosphorylation) exhibit time-dependent dynamic changes postmortem and have the potential to function as “epigenetic molecular clocks” for PMI estimation	H3K27me3 and γ-H2AX modifications demonstrate reproducible patterns of signal decay or elevation across different PMI stages, supporting their utility in molecular timing systems for death estimation [47,102].

**Table 2 genes-16-00940-t002:** Technical limitations, solutions, and implementation challenges.

Limitation	Proposed Solution	Implementation Considerations
Low-Input Sensitivity	Single-cell CUT&Tag [65,66]	High cost; expertise required
Workflow Standardization	Forensic histone consortium; PAT-ChIP validation [156,157]	Cross-laboratory coordination
Modification Stability	Kinetic decay modeling [91,92]	Extensive data collection needed

## Data Availability

Not applicable.

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
