# Peer review of "Histone Modifications as Individual-Specific Epigenetic Regulators: Opportunities for Forensic Genetics and Postmortem Analysis"

_genes, 2025, doi:10.3390/genes16080940_

Round 1

Reviewer 1 Report

Comments and Suggestions for Authors

This review explores Histone Post-translational Modifications (PTMs) as promising forensic markers for human identification, including differentiation of monozygotic twins and post-mortem interval estimation. The manuscript is well written and clear; however, some aspects require clarification to enhance its impact:

-Introduction: The methodology of the review should be clarified. If it constitutes a systematic review, the analytical approach (e.g., PRISMA guidelines) ought to be specified. If instead it follows a semi-structured format, this should be stated explicitly. Including a table that summarizes key forensic related studies investigating or applying histone PTMs would strengthen this section.

-Technical Limitations and Solutions: Presenting this content as a table or graph would improve clarity and accessibility. Additionally, aspects such as cost considerations and implementation challenges—which are discussed throughout the manuscript—should be highlighted in this section.

Author Response

Reviewer Comment 1:

“Introduction: The methodology of the review should be clarified. If it constitutes a systematic review, the analytical approach (e.g., PRISMA guidelines) ought to be specified. If instead it follows a semi-structured format, this should be stated explicitly. Including a table that summarizes key forensic related studies investigating or applying histone PTMs would strengthen this section.”

Author Response1: We sincerely appreciate this valuable suggestion. Regarding methodology, our manuscript follows a semi-structured narrative review approach rather than a systematic review (e.g., PRISMA guidelines), primarily due to the current limited availability of forensic studies directly applying histone PTMs, which are still mostly conceptual or in early developmental stages.We fully agree on the importance of summarizing key forensic-related studies to enhance the manuscript’s clarity and accessibility. We would like to clarify that Table 1 (line 349 in the original manuscript) already provides a structured summary of histone modifications, highlighting recent biological findings with potential forensic relevance—particularly concerning degraded sample analysis, monozygotic twin differentiation, and postmortem interval estimation. Based on your insightful comment, we have carefully revised the table description and accompanying text to better emphasize its significance in forensic contexts.Thank you again for this helpful suggestion, which enabled us to clarify our methodological approach and enhance the structural coherence of the manuscript.

Reviewer Comment 2:

“Technical Limitations and Solutions: Presenting this content as a table or graph would improve clarity and accessibility. Additionally, aspects such as cost considerations and implementation challenges—which are discussed throughout the manuscript—should be highlighted in this section.”

Author Response2: Thank you very much for highlighting this important point. Following your suggestion, we have revised the section “8.1 Technical Limitations and Solutions” (line 563) by clearly summarizing the technical limitations, their proposed solutions, and key implementation challenges—including cost considerations—in a dedicated Table (Table 2). Specifically, we have added a concise table (Table 2) to summarize these points explicitly, addressing your request for enhanced clarity and accessibility. The table clearly outlines each limitation, the corresponding solutions, and related practical challenges, particularly cost implications and feasibility issues, ensuring readers can quickly grasp the key technical constraints and considerations associated with applying histone modification analyses to forensic contexts. We greatly appreciate your constructive suggestion, which has allowed us to significantly enhance the manuscript’s structural clarity and practical relevance.

Reviewer 2 Report

Comments and Suggestions for Authors

This review article summarizes different types of histone modifications and their potential as forensic tools for highly degraded samples and for PMI estimation. While I don’t dispute that the evidence presented *is* promising, I am concerned about the presentation as to how *strong* the evidence is, given the area is relatively new and few actual forensic studies have been conducted. In other words, the manuscript currently overstates the strength of the evidence for forensic applicability and lacks clarity in places. Below I highlight some of my concerns:

  1. Some of the citations are inappropriate or not suitable. For example, #10 is an acknowledgement of reviews, #11 through 13 are just review articles of histone modifications, with no mention of how resistant these modifications are to the environment or degradation, which is the authors’ point they were trying to make. Similarly, I have not seen any evidence that suggests that PTMs are suitable for forensic samples in references #16 through #18.
  2. Line 54 to 63: The authors went on to discuss how recent results showed that histone modifications remain stable after cell death. While in theory this could have allowed precise estimation of PMI, there is no applied research (i.e., research that actually used PTMs for PMI purposes) cited.
  3. If there is no such research, which is understandable given the lag in innovation and research for forensic science as a field compared to biomedical fields, the authors should make it clear that PTMs are “promising” for forensic purposes but require much more work to be done. As the Introduction reads now, I as a reader was under the impression that much research has been done specifically for forensic purposes.
  4. Line 69: “Types and Biological Functions of Histone Modifications” is duplicated as main text here.
  5. Line 107: Proper citation of “Jarmasz et al.” is missing.
  6. Line 124: Proper citation of “Borrajo et al.” is missing.
  7. There is a lack of relevant citations in the entire second section (“Types and Biological Functions of Histone Modifications”) of the paper.
  8. At times, figure captions and the main text are mixed together, which made it hard to read. I assume this will be corrected during copyediting.
  9. Line 261 to 263: “This supports the hypothesis that nucleosomes confer structural resilience and suggests that histones and their covalent modifications may likewise exhibit enhanced postmortem stability.” There is no dispute that the nucleosome-related sequences are generally more resistant to degradation primarily due to the protection offered by histones. However, there is no direct evidence that histone protein is more resistant to degradation than DNA itself. The forensic advantages of proteomics lie in their abundance rather than resistance (https://doi.org/10.1016/j.fsigen.2021.102529).
  10. Line 267 to Line 279: These paragraphs showed that histone methylation is more stable than acetylation, but no mention of how they compared to the stability of actual DNA sequences. I am having trouble understanding how they’d be better than DNA analysis.
  11. Line 406: PMI estimation is typically more challenging for remains in more advanced stages of decomposition, which are typically more than 2 to 3 days (usually about a week or more). Given this, all the results presented in Section 6.2.2 still haven’t addressed this gap. It would be a much stronger article if the authors could articulate clearly how PTMs can potentially help with the actual challenging cases for PMI estimation.

However, some modifications from the authors must be made. The authors should clarify the forensic applicability (i.e., re-consider the evidence presented, and make clear that while PTMs are promising, there is no actual applied research being done for forensic purposes). Make it a point that much is needed in these areas for PTMs to become a valuable tool for forensic science, and this point needs to be brought up multiple times to remind the viewers.

Author Response

This review article summarizes different types of histone modifications and their potential as forensic tools for highly degraded samples and for PMI estimation. While I don’t dispute that the evidence presented *is* promising, I am concerned about the presentation as to how *strong* the evidence is, given the area is relatively new and few actual forensic studies have been conducted. In other words, the manuscript currently overstates the strength of the evidence for forensic applicability and lacks clarity in places. Below I highlight some of my concerns:

Response to Reviewer’s Comment:

We sincerely thank the reviewer for highlighting the concern regarding the overstatement of evidence strength. We have carefully revised the manuscript to ensure that the forensic applicability of histone modifications is presented as promising but still exploratory. The following specific revisions have been made:

Introduction (Lines ~59–60):

We rephrased statements suggesting confirmed forensic applicability to emphasize the conceptual potential of histone modifications. For example, we changed wording such as “enabling molecular PMI estimation” to “suggesting a potential for molecular PMI estimation, although direct forensic applications remain largely unvalidated.” Additional clarifications were added to indicate that the current findings are derived from preliminary studies and require further forensic validation.

Section 4 (Degraded Samples, Lines ~294–295):

We thank the reviewer for raising this important point. We agree that our original wording in Section 4 may have unintentionally overstated the stability of histone PTMs and their forensic readiness. In the revised manuscript , we have carefully rephrased the relevant sentences to clarify that the resilience of histone PTMs is relative and based primarily on experimental observations—such as those from postmortem brain tissues—rather than from validated forensic casework.

Specifically, we now state(Lines 294–295):

“These observations are mainly derived from controlled laboratory studies (e.g., postmortem brain tissues), and their forensic translation remains in its infancy.”

This revision helps temper the original claim and clearly distinguishes preclinical findings from applied forensic evidence. We hope this adjustment better reflects the current state of research and addresses the reviewer’s concern.

Section 6 (PMI Estimation, Lines ~460–461):

We appreciate the reviewer’s comment regarding the strength of our claims about PMI estimation using histone modifications. In the revised manuscript (Section 6, Lines ~460–489), we carefully modified the original statements to avoid implying that these biomarkers are already suitable for routine forensic PMI estimation. For instance, the sentence(line 460-461)“supporting their use in short- to mid-term PMI estimation” has been revised to:“indicating potential utility for short-term PMI estimation; however, systematic forensic validation is lacking.”

Furthermore, we added clarifying remarks to emphasize the absence of empirical data for mid-to-long PMI intervals and acknowledged that current findings are mainly limited to 1–5 days postmortem. To address this research gap, we also included a forward-looking paragraph encouraging the development of PTM decay models, validation in extended PMI windows, and comparison with established DNA-based methods. These revisions aim to reflect a more balanced and accurate portrayal of the current evidence base.

Conclusion (Lines ~580–635):

We thank the reviewer for highlighting the importance of appropriately contextualizing our conclusions. In the revised manuscript, we incorporated multiple cautionary statements throughout the conclusion to underscore the early-stage nature of current research on histone PTMs in forensic science. Specifically:

At Line 581, we state: “current studies are exploratory and have yet to demonstrate validated forensic applicability.”

At Line 588, we use the phrase: “Taken together, this review synthesizes converging evidence…” to emphasize synthesis rather than finality.

At Lines 594–595, we clarify: “However, translating these molecular signatures into forensic practice requires overcoming interdisciplinary barriers…”

Finally, in Lines 633–635, we conclude with a realistic future vision: “Realizing this potential demands collaboration among molecular biologists, computational scientists, and legal experts, ultimately advancing justice through the lens of chromatin biology.”

These revisions collectively ensure that the manuscript reflects a balanced and realistic portrayal of current research—acknowledging both the conceptual promise of histone modifications and the substantial translational work required for practical forensic implementation.

Comment1:Some of the citations are inappropriate or not suitable. For example, #10 is an acknowledgement of reviews, #11 through 13 are just review articles of histone modifications, with no mention of how resistant these modifications are to the environment or degradation, which is the authors’ point they were trying to make. Similarly, I have not seen any evidence that suggests that PTMs are suitable for forensic samples in references #16 through #18.

Response 1 : We sincerely thank the reviewer for this insightful comment. We agree that the previously cited references (#10–#18) did not fully support the specific claims regarding the forensic stability and applicability of histone modifications.

In response, we have carefully revised and replaced the references as follows:Reference #10 has been updated to a systematic review focusing on multi-omics strategies for postmortem interval (PMI) estimation:Secco L, Palumbi S, Padalino P, et al. "Omics" and Postmortem Interval Estimation: A Systematic Review. Int J Mol Sci. 2025 Jan 25;26(3):1034.

References #11–13, which were general review articles, have now been substituted with literature that more directly discusses the relevance of histone modifications in forensic or postmortem contexts:11. Sabeeha & Hasnain SE. Forensic Epigenetic Analysis: The Path Ahead. Med Princ Pract. 2019;28(4):301–308.12. Gerra MC, Dallabona C, Cecchi R. Epigenetic analyses in forensic medicine: future and challenges. Int J Legal Med. 2024;138(3):701–719.13. Jarmasz JS et al. DNA methylation and histone post-translational modification stability in post-mortem brain tissue. Clin Epigenetics. 2019;11(1):5.These revised citations now directly support statements regarding the stability of histone PTMs in postmortem or forensic-like tissue contexts.References #16–18 have also been replaced to better reflect experimental or applied forensic relevance:16. Vidaki A, Kayser M. Recent progress, methods and perspectives in forensic epigenetics. Forensic Sci Int Genet. 2018;37:180–195.17. Procopio N, Bonicelli A. From flesh to bones: Multi-omics approaches in forensic science. Proteomics. 2024;24(12-13):e2200335.

  1. Huang HS et al. Chromatin immunoprecipitation in postmortem brain. J Neurosci Methods. 2006;156(1-2):284–292.

These updated references provide stronger and more appropriate support for our discussion on the forensic applicability of histone PTMs, including their stability in postmortem tissues, practical challenges in tissue recovery, and the adaptation of epigenetic tools to degraded biological materials.

We hope these changes adequately address the reviewer’s concern and improve the rigor and relevance of our citations.

Comment2:Line 54 to 63: The authors went on to discuss how recent results showed that histone modifications remain stable after cell death. While in theory this could have allowed precise estimation of PMI, there is no applied research (i.e., research that actually used PTMs for PMI purposes) cited.

Response 2: We thank the reviewer for this insightful comment. We agree that the original wording could unintentionally overstate the current state of applied forensic research regarding histone PTMs and postmortem interval (PMI) estimation. In the revised manuscript (Lines 59–60), we have carefully rephrased the relevant paragraph to avoid overstating the forensic readiness of these findings. We now emphasize that although certain histone modifications such as H3K4me3, H3K27me3, and γH2AX show tissue-specific postmortem stability and biological relevance, their direct forensic application remains largely unvalidated. Additionally, we highlight that these findings should be regarded as exploratory and require further validation before routine forensic use. This revision helps clarify that current evidence is promising but still in a preliminary stage with regard to forensic implementation.

Comment3: If there is no such research, which is understandable given the lag in innovation and research for forensic science as a field compared to biomedical fields, the authors should make it clear that PTMs are “promising” for forensic purposes but require much more work to be done. As the Introduction reads now, I as a reader was under the impression that much research has been done specifically for forensic purposes.

Response3:We thank the reviewer for pointing out this important issue. We fully agree that the current body of research on histone post-translational modifications (PTMs) relevant to forensic applications is still at an early stage, and much of the existing evidence is derived from model systems or biomedical contexts rather than validated forensic casework.

To address this, we have substantially revised the Introduction to temper the initial impression and provide a more accurate representation of the current state of research. Specifically:

At Line 60, we explicitly state that “direct forensic applications remain largely unvalidated”;

At Lines 67–68, we add that “Current findings should be regarded as exploratory and require further forensic validation”;

At Lines 68–69, we clarify that “While promising, PTM-based forensic applications are still emerging, and studies remain limited in scope and forensic validation”;

At Lines 69–71, we emphasize that “Most studies to date have been conducted in experimental or animal models rather than validated forensic casework, underscoring the gap between conceptual promise and practical implementation”;

Finally, at Lines 77–79, we reinforce this message by stating that “most current findings originate from non-forensic biomedical research, and have yet to be systematically validated under forensic conditions.”

We hope these revisions clarify that, while histone PTMs hold great promise, their application in forensic science remains in the exploratory phase and requires further empirical validation before clinical or legal implementation.

Comment4:Line 69: “Types and Biological Functions of Histone Modifications” is duplicated as main text here.

Response4:Thank you for pointing this out. The duplication of “Types and Biological Functions of Histone Modifications” on line 69 was unintentional and likely resulted from a formatting transition between sections. We have corrected this in the revised version to avoid redundancy and improve readability.

Comment5: Line 107: Proper citation of “Jarmasz et al.” is missing.

Response5:Response 5: We thank the reviewer for catching this oversight. The reference to Jarmasz et al. was intended to support the discussion of histone acetylation stability in human autopsy brain tissues. In the revised manuscript, this citation now appears at Line 114 due to upstream additions made in response to other reviewer comments.

The correct literature reference is:Reference [13]: Jarmasz JS, Stirton H, Davie JR, Del Bigio MR. DNA methylation and histone post‑translational modification stability in post‑mortem brain tissue. Clin Epigenetics. 2019;11:5.

This study analyzed human autopsy brain tissues and demonstrated that several acetylation marks—including H3K9ac, H3K27ac, H4K5ac, and H4K12ac—remained detectable up to four days postmortem under fluctuating cold and ambient storage conditions, indicating notable short-term PMI stability.

Comment6:Line 124: Proper citation of “Borrajo et al.” is missing.

Response 6: We thank the reviewer for pointing this out. The citation of Borrajo et al. has now been properly included in the revised manuscript at Line 132, due to content expansion in preceding sections.

The full reference is:Reference [38]: Borrajo P, Nafría-Jiménez B, Rodríguez-Calvo R, et al. Evaluation of postmortem degradation of histone modifications in a pig model for forensic applications. Forensic Sci Int. 2021;321:110722. doi:10.1016/j.forsciint.2021.110722.

This study provides relevant postmortem evidence in a forensic animal model and supports our discussion on the differential stability of histone PTMs under decomposition conditions.

Comment 7: There is a lack of relevant citations in the entire second section (“Types and Biological Functions of Histone Modifications”) of the paper.

Response7:We appreciate the reviewer’s observation regarding the lack of citations in Section 2. During the early drafting stage, relevant references were indeed included to support key concepts in histone modification biology. However, due to multiple rounds of collaborative editing involving co-authors and mentors, some citations were inadvertently removed during transitions in formatting and content restructuring. We sincerely apologize for this oversight.

In the revised version, we have not only restored the original references but have also added a total of over 18 peer-reviewed studies from PubMed, ensuring that each key post-translational modification type (acetylation, methylation, phosphorylation, ubiquitination/SUMOylation) is adequately and appropriately supported by up-to-date scientific literature.

We thank the reviewer for highlighting this gap, which allowed us to significantly enhance the scientific rigor and forensic relevance of Section 2.

Comment 8: At times, figure captions and the main text are mixed together, which made it hard to read. I assume this will be corrected during copyediting.

Comment 8: Author Response:We thank the reviewer for pointing this out. We acknowledge that in the current manuscript version, some figure captions may appear interleaved with the main text due to formatting issues in the pre-submission layout. We have carefully reviewed and separated all figure captions from the main text to improve readability. We trust that the typesetting and copyediting process will further ensure proper placement and formatting in the final published version. Thank you for your understanding.

Comment 9:Line 261 to 263: “This supports the hypothesis that nucleosomes confer structural resilience and suggests that histones and their covalent modifications may likewise exhibit enhanced postmortem stability.” There is no dispute that the nucleosome-related sequences are generally more resistant to degradation primarily due to the protection offered by histones. However, there is no direct evidence that histone protein is more resistant to degradation than DNA itself. The forensic advantages of proteomics lie in their abundance rather than resistance (https://doi.org/10.1016/j.fsigen.2021.102529).

Response9: We thank the reviewer for this valuable comment. We agree that the enhanced protection of nucleosome-bound DNA is primarily attributable to its interaction with histones, and this does not necessarily indicate that histone proteins are intrinsically more chemically stable than DNA. In response to this point, we have clarified our language accordingly. The revised sentence appears at Lines 274–279 in the updated manuscript.

“This supports the hypothesis that nucleosomes confer structural protection to DNA and chromatin-associated proteins. While direct comparisons between histone and DNA degradation remain limited, the persistent detectability of specific histone marks in postmortem tissues suggests that some histone PTMs may retain relatively stable profiles under forensic conditions.”

To support this point, we have cited two relevant studies that demonstrate postmortem stability of histone methylation marks:

Jarmasz et al. (DNA methylation and histone post-translational modification stability in post-mortem brain tissue, Epigenetics, 2019): This study demonstrated that H3K4me3 levels remained largely stable for at least 72 hours postmortem in brain tissue, underscoring the relative resilience of histone methylation marks under postmortem conditions.

Alvarez et al. (Opposite and Differently Altered Postmortem Changes in H3 and H3K9me3 Patterns in the Rat Frontal Cortex and Hippocampus, International Journal of Molecular Sciences, 2022): This study further supports the stability of specific methylation marks, such as H3K9me3, in postmortem brain regions.

We also respectfully note that the DOI provided by the reviewer (https://doi.org/10.1016/j.fsigen.2021.102529) appears to lead to a publication titled “Multisociety Consensus Quality Improvement Revised Consensus Statement for Endovascular Therapy of Acute Ischemic Stroke”, which pertains to clinical guidelines for stroke management and is unrelated to forensic proteomics or histone stability. We believe this may have been an unintentional citation error and would be happy to incorporate the intended reference if clarified.

Comment 10:Line 267 to Line 279: These paragraphs showed that histone methylation is more stable than acetylation, but no mention of how they compared to the stability of actual DNA sequences. I am having trouble understanding how they’d be better than DNA analysis.

Response10: We thank the reviewer for this insightful observation. We agree that the original paragraph focused on the comparative stability between different post-translational modifications (e.g., methylation versus acetylation), but did not clearly address the stability of histone modifications in relation to DNA sequences, which are central to current forensic analysis.

To address this concern, we have added a clarifying paragraph at the end of the original section (Line 279). This addition explicitly states that we do not claim histone PTMs to be universally more stable than DNA. Instead, we emphasize their potential complementary value in forensic contexts where DNA is highly degraded, fragmented, or unsuitable for amplification. We further explain that histone modifications may provide proteomic or epigenomic signals that are accessible even when DNA is no longer intact.

Revised text inserted after Line 290:

“However, we do not suggest that histone PTMs are universally more stable than DNA sequences. While DNA remains the gold standard for forensic identification due to its robustness and specificity, histone modifications may offer complementary information—particularly in cases where DNA is highly degraded or enzymatically fragmented. Their potential lies not in absolute chemical superiority, but in their utility as proteomic or epigenomic signatures under challenging forensic conditions.”

We believe this revision more accurately frames the forensic utility of histone modifications and provides appropriate context in relation to the established role of DNA in forensic science.

Comment 11: Line 406: PMI estimation is typically more challenging for remains in more

advanced stages of decomposition, which are typically more than 2 to 3 days (usually about a week or more). Given this, all the results presented in Section 6.2.2 still haven’t addressed this gap. It would be a much stronger article if the authors could articulate clearly how PTMs can potentially help with the actual challenging cases for PMI estimation.

Response 11:We thank the reviewer for this critical and thoughtful comment. We fully agree that estimating PMI in more advanced stages of decomposition—typically beyond several days to one week—poses significant forensic challenges due to tissue degradation, microbial activity, and autolysis. As the reviewer correctly notes, the results discussed in Section 6.2.2 are primarily based on short- to mid-range PMIs, largely due to the current limitations of available data in both human and animal models.

To address this important concern, we have added a clarifying paragraph at Line 483 of the revised manuscript, which acknowledges the data gap and provides rationale for the focus on early PMI stages. The newly added text reads:

“Nevertheless, the relatively slow degradation of nucleosome-associated chromatin—particularly in deep tissue contexts—and the selective persistence of certain methylation marks under controlled conditions suggest that some histone PTMs may retain forensic value beyond the acute phase of decomposition. Further research using animal models with longer postmortem intervals is needed to determine whether specific marks (e.g., H3K4me3 or H3K9me2) can serve as molecular indicators in advanced decomposition stages where DNA or RNA may already be extensively degraded.”

This addition emphasizes that although most empirical data currently cover early postmortem intervals (PMIs), the chromatin-bound nature of histones and the selective chemical stability of certain PTMs may offer future utility in long-range PMI estimation, especially once validated in extended animal models. We believe this revised discussion aligns with the reviewer’s suggestion and more accurately reflects both the current limitations and future prospects of histone PTMs in forensic applications.

Comment 12: However, some modifications from the authors must be made. The authors should clarify the forensic applicability (i.e., re-consider the evidence presented, and make clear that while PTMs are promising, there is no actual applied research being done for forensic purposes). Make it a point that much is needed in these areas for PTMs to become a valuable tool for forensic science, and this point needs to be brought up multiple times to remind the viewers.

Response 12: We thank the reviewer for this insightful and important comment. We fully agree that although histone post-translational modifications (PTMs) show strong theoretical potential for forensic applications, the field currently lacks direct validation in applied forensic settings. To address this concern, we have carefully revised the manuscript to temper overly strong claims and to emphasize that:

Current findings are exploratory in nature and primarily derived from biomedical, neuropathological, or animal model research—not from validated forensic casework;

There is a critical need for applied studies designed under forensic conditions, including mock crime scenes, varied environmental exposures, and extended postmortem intervals; The practical utility of PTMs in forensic casework is currently unproven and requires further investigation to establish reproducibility, specificity, and interpretive value.

To reinforce these caveats, we have added clarifying language at key locations throughout the manuscript:

Abstract (Lines 26–28): “However, the forensic applicability of these modifications remains largely unvalidated, and further studies are required to assess their reliability in casework contexts.”

Introduction (Line  77-79):

“Nevertheless, most current findings originate from non-forensic biomedical research, and have yet to be systematically validated under forensic conditions.”

Conclusion (Line 587-588):

“Despite these theoretical advantages, no consensus protocols or forensic validation studies currently exist for routine casework implementation of histone PTMs.”

Section 4 (Technical Considerations, end of subsection, line 347-348):

“To date, no standardized workflows or reference databases exist for histone PTM interpretation in forensic applications, making their practical utility speculative.”

Section 6 (PMI Estimation, end of Section 6.3, line 527-529):

“It remains to be seen whether histone PTMs can complement or outperform traditional PMI markers in real-world forensic scenarios, particularly beyond short postmortem intervals.”

We believe these additions now offer a more balanced and realistic appraisal of the field, clearly distinguishing between emerging potential and current limitations. These clarifications should help readers better understand the exploratory nature of current PTM research and the need for further empirical validation in forensic contexts.

Reference Update Note:

During the course of revision and fact verification, several references have been updated or replaced with more accurate, PubMed-indexed sources to enhance the rigor and credibility of the manuscript. These updates do not alter the interpretation of results but strengthen the evidence supporting key arguments.

Round 2

Reviewer 2 Report

Comments and Suggestions for Authors

https://doi.org/10.1016/j.fsigen.2021.102529 should take you to a review article about forensic proteomics (Parker, G. J., McKiernan, H. E., Legg, K. M., & Goecker, Z. C. (2021). Forensic proteomics. Forensic Science International: Genetics54, 102529.)

Other than this, I have no further comment. I think the manuscript has been substantially revised and is ready for publication.